# Assessment of miR-98-5p, miR-152-3p, miR-326 and miR-4289 Expression as Biomarker for Prostate Cancer Diagnosis

**DOI:** 10.3390/ijms20051154

**Published:** 2019-03-06

**Authors:** Leire Moya, Jonelle Meijer, Sarah Schubert, Farhana Matin, Jyotsna Batra

**Affiliations:** 1Australian Prostate Cancer Research Centre—Queensland, Translational Research Institute, 37 Kent St, Brisbane, Queensland 4102, Australia; leire.moya@qut.edu.au (L.M.); jonellemeijer@hotmail.com (J.M.); sarah.schubert@connect.qut.edu.au (S.S.); farhana.matin@qut.edu.au (F.M.); 2Cancer Program, School of Biomedical Sciences, Institute of Health and Biomedical Innovation, Queensland University of Technology, 60 Musk Avenue, Brisbane, Queensland 4059, Australia

**Keywords:** miRNAs, biomarker, prostate cancer, diagnosis, prognosis

## Abstract

Prostate cancer (PCa) is one of the most commonly diagnosed cancers worldwide, accounting for almost 1 in 5 new cancer diagnoses in the US alone. The current non-invasive biomarker prostate specific antigen (PSA) has lately been presented with many limitations, such as low specificity and often associated with over-diagnosis. The dysregulation of miRNAs in cancer has been widely reported and it has often been shown to be specific, sensitive and stable, suggesting miRNAs could be a potential specific biomarker for the disease. Previously, we identified four miRNAs that are significantly upregulated in plasma from PCa patients when compared to healthy controls: miR-98-5p, miR-152-3p, miR-326 and miR-4289. This panel showed high specificity and sensitivity in detecting PCa (area under the curve (AUC) = 0.88). To investigate the specificity of these miRNAs as biomarkers for PCa, we undertook an in depth analysis on these miRNAs in cancer from the existing literature and data. Additionally, we explored their prognostic value found in the literature when available. Most studies showed these miRNAs are downregulated in cancer and this is often associated with cancer progression and poorer overall survival rate. These results suggest our four miRNA signatures could potentially become a specific PCa diagnostic tool of which prognostic potential should also be explored.

## 1. Introduction

Prostate cancer (PCa) is the second most commonly diagnosed cancer worldwide with more than 1.1 million new cancer cases each year, accounting for 15% of all cancers diagnosed [1]. Despite this high incidence, the most commonly used molecular method of detection and monitoring of the disease recurrence is the prostate specific antigen (PSA) test, discovered 40 years ago [2]. Although this method was initially thought to improve the early diagnosis of the disease [3], it has been criticized over the last decade due to its lack of specificity, giving false positives and often leading to over-diagnoses [4,5]. Notably, the American Urological Association recommended against PSA screening for PCa and encouraged the identification of novel markers that can identify men at greater risk of developing and progressing on the disease [6]. Furthermore, the U.S. Preventative Services Task force also recommended against PSA screening in men over 70 years old [7] and, in Australia, the Cancer Council Australia has recently reported that some men with PCa have normal PSA levels while only one in three men with higher PSA levels has the significant disease [8]. In this light, novel and more specific diagnostic biomarkers are required for this disease. Highly specific and sensitive biomarkers will not only improve the quality of life of many men worldwide by reducing unnecessary over-diagnosis related anxiety and depression [9] but they could also alleviate some of the economic burden from unnecessarily high cost treatments [10].

MicroRNAs (miRNAs) are a class of small, non-coding RNA molecules. They bind to their target mRNAs to regulate gene expression [11] and some in silico and functional analysis estimated that over half of the genes that code for human proteins are regulated by miRNAs [12]. Cancer development involves normal cells experiencing genetic instability [13] and this genomic instability has a great impact on the miRNAs’ expression profile by affecting their genomic loci or processing and, in consequence, by altering the critical cellular pathways that they regulate such as cell proliferation, differentiation and apoptosis, contributing to cancer initiation and progression [14]. Indeed, dysregulation of miRNAs has been reported in many cancers [15,16,17,18,19,20,21] and the so-called miRNA signatures have proved to be accurate tools not only to detect different cancer types [19,22,23] but also for cancer staging [24,25]. This is highly relevant to allow for an accurate diagnosis, early detection and implementation of the most appropriate treatment, all of which can potentially improve prognosis.

We have recently identified a panel of four miRNAs: miR-98-5p, miR-152-3p, miR-326 and miR-4289 [26] with elevated levels in plasma samples from PCa patients when compared to healthy controls. This signature was validated in two independent cohorts. Furthermore, when combined, the diagnostic power of our signature as represented by the area under the curve (AUC) = 0.88, proved to be greater than that previously reported for PSA [27,28]. To further determine the potential of this novel signature in PCa and its specificity in this disease, an in depth systematic analysis was carried out, where more than 200 research articles on these four miRNAs in cancers in the last decade have been revisited. Expression dysregulation in cancer, survival data and disease association analysis have been gathered for the four miRNAs. We have summarised the most recent reported dysregulation data for these four miRNAs in cancer and determined whether the upregulation observed by us in PCa is specific for this disease. Additionally, we also considered their prognosis significance when available in the literature. 

## 2. Results 

### 2.1. Literature Retrieval

More than 300 abstracts were retrieved initially from PubMed. These included information for the four miRNAs of interest related to their dysregulation in cancer and/or survival. Those studies with less than 20 patients were discarded, leaving ~200 abstracts for further inspection. Papers with no clinical and pathological data and those that focused on the basic biology of the miRNA were excluded. This left 78 studies to be examined in further detail to extract potential data. Of these, 38 research articles reported significant results for the miRNAs of interest for their association with one or more forms of cancer, from which data was subsequently extracted and presented in this article.

### 2.2. Dysregulation of miR-98-5p, miR-152-3p, miR-326 and miR-4289 in Cancer Patient Samples

The main focus of this section was to compile recent reported dysregulation expression data for our miRNA signature in all cancers and assess its specificity for PCa. Cancer-related dysregulation that met our selection criteria was found for three of the miRNAs, with no reported data for miR-4289. However, two examples have been included at the end of this section to get a preliminary appreciation of what is available for miR-4289 (Table 1). Additionally, survival data was also collected when available. Some studies reported hazard ratios (HR) values while others associated miRNA expression levels with survival rate (Table 2).

#### 2.2.1. Dysregulation of miR-98-5p in Cancer

Our analysis showed a higher plasma expression of miR-98-5p is associated with an increase in the likelihood of developing PCa (cohort 1: β = 1.75, 95% confidence interval (CI) = 0.36–3.14, P = 0.056 and cohort 2: β = 1.80, 95% CI = 1.20–2.40, P = 1.47 × 10^−8^) [26]. In a recent meta-analysis conducted by Pashaei et al. on six independent datasets that identified miRNAs associated with PCa, a significant (P < 0.05) upregulation of miR-98-5p in recurrent vs. non-recurrent PCa patients after radical prostatectomy surgery was observed [29]. These two studies suggest that miR-98-5p upregulation in plasma and/or in prostate tissue could be a diagnostic biomarker for this disease and also it could serve as a recurrence biomarker (Table 1).

By contrast, miR-98-5p was downregulated in lung cancer tissue when compared to adjacent cancer-free tissue (*N* = 26, P < 0.01) [34] and in the serum of lung cancer patients (*N* = 127) when compared to controls (*N* = 60, P < 0.05) [35]. Additionally, Wang et al. observed this downregulation was positively correlated with lymph node metastasis, worse TNM (Tumor, Nodes and Metastasis) stage and decrease in the overall survival of patients [35] (Table 1). Similarly, miR-98-5p was reported to be downregulated in melanoma patient tissue samples (*N* = 24) when compared to cancer-free controls (*N* = 24, P < 0.01) [36] and in metastatic melanoma (*N* = 15) when compared with primary tissue (*N* = 15, P < 0.001) [36]. A lower expression was associated with a later T staging and a worse prognosis (P < 0.001) [36] (Table 1). The downregulation of miR-98-5p was also observed in hepatocellular carcinoma (HCC) tissue when compared to adjacent cancer-free cells in two studies carried out by Zhou et al. (*N* = 144, P < 0.001) [30] and Wang et al. (*N* = 30, P < 0.05) [31]. This downregulation was significantly associated with larger tumour size (P = 0.025), lower percentage survival (P = 0.0125) and metastasis (P = 0.016) [30].

In addition, miR-98-5p was also observed to be downregulated in esophageal squamous cell carcinoma (ESCC) by Huang et al. They reported a significant downregulation of miR-98-5p in ESCC tumor tissue when compared to adjacent cancer-free cells (*N* = 40, P < 0.01) [32] and this lower expression was significantly correlated with a higher pathological grade (P = 0.004), later tumor stage (P = 0.003) and lymph node metastasis (P = 0.004) (Table 1). Furthermore, Liu et al. also observed a consistent downregulation of miR-98-5p in nasopharyngeal carcinoma (NPC) tumor tissue when compared to adjacent cancer-free cells (*N* = 30, P < 0.05) (Table 1). Finally, Chen et al. found miR-98-5p expression was significantly lower in brain cancerous tissue when compared to adjacent non-cancerous tissue (*N* = 26, P = 0.0165) [33] (Table 1).

Survival data for miR-98-5p was reported for lung, melanoma, HCC and ESCC, where high miRNA expression levels in tissue significantly (P < 0.05) correlated with higher survival rates for the first three cancers and a lower metastasis incidence for all of them [30,32,35,36] (Table 2).

In summary, the current reported expression data for miR-98-5p shows this miRNA is downregulated in lung, melanoma, HCC, osteosarcoma (OSC), and NPC tumor tissue samples when compared to adjacent non-malignant or healthy tissues. This downregulation may contribute to cancer progression and development by affecting several pathways and genes such as those summarised in Table 2. By contrast, an upregulation of miR-98-5p in plasma samples from PCa patients when compared to controls has been observed [26] and this upregulation has also been reported in recurred prostate tumors [29]. These results suggest that an upregulation of miR-98-5p is highly specific to PCa. However, in clinical settings, low levels of miR-98-5p should be interpreted cautiously as that might indicate malignancy of other organs.

#### 2.2.2. Dysregulation of miR-152-3p in Cancer

Our study significantly correlated higher expression of plasma miR-152-3p in PCa patients when compared to healthy controls in two cohorts (cohort 1: *N* = 61, β = 1.89, 95% CI = 0.99–2.79, P = 1.4 × 10^−4^ and cohort 2: *N* = 58, β = 3.38, 95% CI = 2.27−4.49, P = 8.94 × 10^−9^) [26] (Table 1). In contrast, miR-152-3p has been described as a possible tumor suppressor in numerous studies as its downregulation has been reported in several cancer tissues. For example, in PCa, two independent studies reported a downregulation of miR-152-3p when comparing PCa tissue (*N* = 48) vs. non-malignant tissue (*N* = 15, P < 0.05) [39] and PCa tissue vs. adjacent cancer-free cells (*N* = 97, P < 0.0001) [50] and primary tissue (*N* = 97) vs. metastatic PCa samples (*N* = 13, P < 0.001) [50] (Table 1). Additionally, a lower miR-152-3p expression was associated with higher Gleason scores (>7), a more advanced pathological T stage [39] and a higher biochemical recurrence-free survival (P = 0.0004) [50]. Similarly, Kristensen et al. also observed a downregulation of miR-152-3p in PCa (*N* = 134) when compared to benign prostatic hyperplasia (BPH) patients (*N* = 13, P = 0.00004) and these results were validated in their second cohort of patients (*N* = 138) and BPH controls (*N* = 19, P = 0.0003). In line with these studies, our analysis of TCGA (The Cancer Genome Atlas) data revealed that miR-152-3p expression was significantly (P = 0.0011) lower in prostate tumor tissues (*N* = 181) when compared to non-malignant prostate tissues (*N* = 50) [26] (Table 1).

Similarly, miR-152-3p has been reported to be downregulated in colorectal cancer (CRC) tissue when compared to adjacent cancer-free cells in three independent studies by Li et al. (*N* = 28, P < 0.01) [49], Chen et al. (*N* = 101, P < 0.01) [50] and Wang et al. (*N* = 202, P < 0.001) [51]. It was also observed that a low miR-152-3p expression was significantly correlated with some clinic-pathological features such as larger tumor size (P = 0.004) and advanced tumor staging (P = 0.002) [50]. Interestingly, Chen et al. also reported an upregulation of miR-152-3p in CRC plasma samples (*N* = 31) when compared to healthy controls (*N* = 52, P = 0.02) [46] (Table 1), an observation similar to PCa.

Downregulation of miR-152-3p in gastric cancer tissue when compared to normal adjacent tissue was observed by Zhai et al. (*N* = 30, P < 0.05) [55] and Chen et al. (*N* = 101, P = 0.038) [50]. As seen previously, a low miR-152-3p expression significantly correlated with a larger tumor size (P = 0.023) and a more advanced stage (P = 0.018) [50] (Table 1). Furthermore, miR-152-3p also showed a significant (P < 0.05) downregulation in ovarian cancer tissue (*N* = 78) when compared to healthy ovarian tissue (*N* = 17, P < 0.05) [54].

Two independent studies by Xu et al. (*N* = 22, P < 0.05) [47] and Maimaitiming et al. (*N* = 32, P < 0.05) [48] observed a downregulation of miR-152-3p in breast tumor tissue when compared to adjacent normal breast tissue. Contrary to the observations reported in breast cancer tissue, Chen et al. observed an upregulation of miR-152 in breast cancer plasma samples (*N* = 53) when compared to healthy controls (*N* = 49, P = 0.003) [46] (Table 1).

In glioma tissue, the miR-152 was also downregulated (*N* = 20, P < 0.001) when compared to adjacent normal cells by Zhang et al. [43] (Table 1). Additionally, miR-152-3p was also shown to be downregulated in HCC in two different analysis: HCC tissue vs. adjacent normal cells (*N* = 89, P < 0.01) [41] and HCC tissue (*N* = 55) vs. healthy controls (*N* = 76, P < 0.001) [42]. As seen in other cancers, this downregulation was associated with later cancer staging (P = 0.013) and larger tumor size (P = 0.037) [42] (Table 1).

Dysregulation of miR-152-3p was reported in lung cancer where two independent studies done with plasma samples gave opposite results. In the first study by Dou et al., miR-152-3p plasma levels were reported to be downregulated in lung cancer samples (*N* = 120) when compared to plasma from healthy controls (*N* = 360, P = 0.014) [45]. By contrast, a second study conducted by Chen et al. reported an upregulation of miR-152-3p in plasma samples from lung cancer patients (*N* = 55) when compared to healthy controls (*N* = 53, P = 0.00015) [46]. In a third study in which lung tissue samples were analysed, a lower expression of miR-152-3p was observed in lung cancer patient tissue when compared to adjacent cancer-free cells (*N* = 36, P < 0.05) [44]. Although none of these studies associated miR-152-3p dysregulation with a larger tumor size, it correlated with other clinicopathological features such as poor differentiation status (P = 0.001) and later staging (P < 0.001) [47] (Table 1).

As the inverse dysregulation between plasma/serum and tissue has been previously mentioned for prostate, CRC, breast and lung cancers, in bladder cancer similar results were reported. Jiang et al. observed an upregulation of miR-152 in serum of bladder cancer patients (*N* = 250) when compared to controls (*N* = 240, P < 0.05) [52] while a downregulation in tissue samples from patients with bladder cancer (*N* = 238) when compared to healthy controls (*N* = 121, P < 0.00001) [53] was reported (Table 1).

Reported survival data from HCC serum samples (*N* = 76) showed lower miR-152 expression levels correlated with shorter overall survival in a univariate analysis (HR = 2.53, 95% CI = 1.22−5.22, P = 0.0012) [41]. This result did not, however, reach significance in the multivariate analysis (HR = 1.73, 95% CI = 0.83−3.6, P = 0.141), suggesting miR-152 cannot be used as an independent prognostic marker in HCC.

Survival analysis in PCa showed that the higher tissue expression levels correlated with a better prognosis and a lower metastatic incidence [40] (Table 2). This suggests miR-152 has the potential to be a specific prognosis biomarker for PCa.

In summary, miR-152 has been reported to be downregulated in prostate, CRC, gastric, ovarian, breast, glioma, and HCC, lung and bladder tissue when compared to adjacent and/or healthy tissue. This downregulation is often associated with cancer development and progression. By contrast, an upregulation has been observed in prostate, lung, breast, CRC and bladder serum/plasma patient samples when compared to healthy samples. Contradictory results have been reported for plasma breast cancer samples, where both up- and downregulation have been observed. Further studies with bigger cohorts are required to determine the directionality of such dysregulations. Nevertheless, from these data, the upregulation of miR-152 expression in plasma was not found to be a specific biomarker for PCa when analysed alone, but could also suggest lung, breast, CRC or bladder cancers.

#### 2.2.3. Dysregulation of miR-326 in Cancer

In our study, the β values obtained from the regression coefficient analysis showed high miR-326 plasma levels in PCa patients was associated with a higher likelihood of developing this disease (cohort 1: *N* = 61, β = 6.18, 95% CI = 4.52–7.83, P = 9.15 × 10^−13^ and cohort 2: *N* = 58, β = 10.42, 95% CI = 8.09–12.76, P = 0.00E +^00^) [26] (Table 1).

By contrast, miR-326 was reported to be downregulated by Li et al. in gastric cancer [60] when comparing gastric tumor tissue with adjacent cancer free tissue (*N* = 136, P < 0.001). This low expression also significantly correlated with some clinic-pathological characteristics such as advanced clinical staging (P = 0.003), tumour depth (P = 0.026), both lymph node (P = 0.004) and distant metastasis (P = 0.037) and a poorer survival rate (P < 0.01) [60] (Table 1).

Downregulation of miR-326 was also reported in lung cancer when cancerous lung tissue was compared to adjacent healthy cells (*N* = 39, P < 0.05) and this was significantly associated with a poorer prognosis (P = 0.0061) [56] (Table 1).

Again, miR-326 was reported to be downregulated in CRC by Wu et al. [61] in CRC tissue when compared to adjacent normal tissue (*N* = 114, P < 0.05) and the lower expression was also associated with an increase in metastasis, disease recurrence (P < 0.05) and a poorer survival rate (P = 0.018) (Table 1).

Similarly, miR-326 was also observed to be downregulated in OSC by Cao et al. [61], where a significant downregulation was reported in both OSC tissue patients vs. normal tissue (*N* = 30, P<0.05) and OSC tissue vs. adjacent cancer free tissue (*N* = 6, P < 0.05). In this study a significant decrease in expression was also observed in patient serum samples when compared to healthy controls (*N* = 60, P < 0.05). Additionally, lower miR-326 expression was associated with a poorer survival rate (P < 0.05), a higher metastatic incidence and a more advanced clinical stage (P < 0.05) without affecting the tumor size [61] (Table 1).

Finally, both Zhang et al. (*N* = 25, P = 0.0109) [58] and Wu et al. (*N* = 30, P < 0.01) [59] observed a downregulation of miR-326 in cervical cancer tissue when compared to healthy control tissue (Table 1).

The reported survival data found for miR-326 is summarised in Table 2. Data was found for the following cancers: prostate, CRC, glioma (including glioblastoma, GBM), OSC, gastric and lung cancers. In PCa, Kristensen et al. observed that low miR-326 expression in patient tissue was associated with early biochemical recurrence in two cohorts (cohort 1: *N* = 126, HR = 0.90, 95% CI = 0.82–0.99, P = 0.032 and cohort 2: *N* = 110, HR = 0.91, 95% CI = 0.84–0.99, P = 0.023) [38]. However, these results could not be validated in a third cohort, where an opposite HR directionality was observed and it did not reach significance (*N* = 99, HR = 1.46, 95% CI = 0.76–0.2.83, P = 0.256). Also, it did not remain significant in their multivariate analysis, suggesting the results observed are not independent of the covariates tested (PSA, pT stage, Gleason scores and margin status). In CRC, Wu et al. reported a significant association of high miR-326 CRC tissue expression levels with a better overall survival (*N* = 114, HR = 0.58, 95% CI = 0.28–0.79, P = 0.011) and progression-free survival (HR = 0.811, 95% CI = 0.418–0.951, P = 0.017) and these were independent of clinical and pathological characteristics such as age, gender, tumor location, TNM stage and differentiation status [57] (Table 2). It was also associated with a lower metastatic incidence.

Qiu et al. showed higher expression of miR-326 in GBM tissue was associated with a better prognostic and this was independent of other factors such as age, gender and recurrence (*N* = 458, HR = 0.70, 95% CI = 0.54–0.90, P = 0.006) [62] (Table 2).

Similarly, low miR-326 expression in glioma tissue showed to be an independent marker of poor overall survival prognosis (*N* = 108, HR = 6.5, 95% CI = 1.20–19.70, P < 0.001) [63]. Table 2. Furthermore, low miR-326 glioma tissue expression was also significantly associated with poor overall survival in patients with high pathological grades (III–IV: P < 0.001), but significance was not reached when the grades were low (I–II: P = 0.07) [63]. In OSC a lower miR-326 serum expression was associated with a lower metastasis incidence and it was an independent overall survival prognostic marker (*N* = 60, HR = 3.9, 95% CI = 1.13–12.35, P = 0.001) [61] (Table 2). In gastric cancer, low miR-326 expression was also significantly associated with an overall poorer prognostic (*N* = 136, HR = 1.51, 95% CI = 1.08–2.76, P = 0.02) [60] and their inverse are shown in Table 2. A higher miR-326 expression tissue level was also associated with a lower metastatic rate. Finally, in lung cancer tissue higher miR-326 tissue expression levels were associated with a higher survival rate [56] (Table 2).

To summarise, miR-326 has been shown to be downregulated in gastric, lung, CRC, ESCC and cervical tumor tissue when compared to adjacent/healthy tissue and this is often associated with cancer progression, disease recurrence and poorer prognosis. This downregulation was also observed in serum ESCC patient samples when compared to controls. By contrast, we have observed an upregulation of this miRNA in PCa plasma samples when compared to controls. Additionally, tumor miR-326 overexpression is associated with a better overall survival in prostate, CRC, GBM, glioma, OSC, gastric and lung cancers and a lower metastasis incidence. High levels of miR-326 in PCa tissue samples when compared to healthy controls also showed to be associated with disease recurrence although it was not independent from other clinical factors such as PSA, staging and Gleason score. Overall, all these results suggest this miRNA is also a good specific candidate biomarker for PCa at the diagnostic and prognostic levels.

#### 2.2.4. Dysregulation of miR-4289 in Cancer

Higher plasma levels were observed in PCa patient samples when compared to controls in our two cohorts as shown by univariate analysis (Table 1) [26]. The first cohort associated higher plasma levels with an increased risk of PCa as determined by B (*N* = 61, β = 0.83, 95% CI = 0.5–1.16, P = 3.11 × 10^−6^). Similarly, the second cohort also associated elevated plasma expression levels with a higher risk of developing the disease (*N* = 58, β = 5.71, 95%CI = 4.18–7.24, P = 8.80 × 10^−13^) [26].

The research done on miR-4289 is currently very limited, with only a few studies reporting its role in cancer. Because of this availability shortage, two studies are here mentioned despite of not fitting the search criteria applied. Dong et al. undertook miRNA profiling in three samples only to detect differentially expressed miRNAs between GBM patients and healthy controls. From the 752 miRNAs analysed, 115 miRNAs were found to be upregulated being miR-4289 one of them (P = 0.0002) [64]. The second study, used a stratified cohort based on the expression of two particular genes, the serine-threonine protein kinase AKT1 and AKT2 [65]. This study investigated the role of miRNAs as effectors of increasing the AKT1 and AKT2 gene numbers within lung carcinomas. miR-4289 was found to be upregulated in two of the stratified lung tissue cancer cohorts (AKT1+ and AKT2+).

No reported survival data in cancer was found for the miR-4289.

### 2.3. Differential Expression of the miRNA Signature in Cancerous Samples when Compared to Non-Cancerous Samples from Microarray Dataset

Differential expression data reported in dbDEMC 2.0 was found for three out of the four miRNAs (Figure 1) and all the miRNAs were significantly (P < 0.05) dysregulated in cancer samples when compared to controls. Some experiments had few of the miRNAs duplicated and these were discarded, leaving only one value. Differential expression values were found in many datasets for miR-98-5p (*N* = 48), miR-152-3p (*N* = 49) and miR-326 (*N* = 49) and are presented in Figure 1. Data for miR-4289 was found only for three cancers (biliary tract, CRC and gastric cancers) from two different experiments. In the three cases, miR-4289 was significantly downregulated.

Additional information such as the experiment ID, Gene Expression Omnibus (GEO)/TCGA information, cancer subtypes, T values, P values and β values and adjusted P values are all summarised in Appendix A. For those cancers with a limited number of values, a grouping system was followed. Head and neck included: head and neck, NSP, tonsil, thyroid, oral squamous and ESCC cancers. Liver is comprised of both liver and HCC cancers. Colon and colorectal cancers were combined. Sarcoma included both sarcomas synovial and liposarcoma, and endometrial was grouped alongside uterus cancer. From this data, none of the miRNAs showed to be a specific biomarker for any cancer studied (Figure 1). Non-parametric, univariate analysis of variance (ANOVA) revealed significant differences in between cancers (P = 0.035) only for miR-326.

### 2.4. Functional Role of the miRNA Signature in Cancer

The literature research previously presented also provided a number of validated targets for the different miRNAs that compose the signature. The method of validation, gene target effect and involved pathway or molecular function are summarised in Table 3. Briefly, five target genes and one positive regulator were reported for miR-98 in lung, melanoma, HCC, brain and NPC cancers. Six target genes were validated for miR-152 in gastric, ovarian, breast and lung cancers and five target genes, plus two negative regulators, were confirmed for miR-326 in gastric, lung, glioma, OSC and cervical cancers (Table 3). Our previous in silico analysis of miRNA target genes generated 1055 targets of these four miRNAs implicating them in over 250 pathways [26]. The maximum number of targets were found to be involved in cancer pathways including viral carcinogenesis, bladder cancer, chronic myeloid leukemia, PCa, non-small cell lung cancer, basal cell carcinoma, and glioma (P ≤ 0.05). Identifying miRNA target genes and pathways to understand the molecular basis of cancer pathogenesis is challenging because there are several direct and indirect targets and pathways that drive cancer. For example, the insulin signalling pathway consisting of insulin growth factor receptors (IGFRs) are important mediators of cancer. Similarly, cyclin dependent kinases (CDKs) targeted by our identified miRNAs play a vital role in cell cycle progression which, in turn, is an important pathway in cancer.

## 3. Discussion

A good usable biomarker has to be selective and objectively measured to evaluate diverse biological or pathogenic processes, as described by the Biomarker Consortium group [71]. Amongst others, a biomarker can assist at different levels in managing a certain disease. For instance, it can be applied to differentiate between disease and healthy status, assist in staging and classifying the extent of a disease, improve the prognosis and/or monitoring the clinical response to a potential treatment [71]. miRNAs have extensively been reported to be dysregulated in cancer [72,73,74,75] and they have been described as highly specific, sensitive and stable molecules, which makes them potential fingerprint tools to diagnose a specific cancer and its progression [72]. Recently, some studies have shown this tumor-specificity in different fluids such as saliva for oral cancer [76], urine for bladder cancer [77] and plasma for lung, CRC and breast cancers [46]. As mentioned before, the main PCa biomarker used today, although sensitive, can lead to over-diagnosis due to a relative low specificity [4,5]. We have recently identified a plasma miRNA signature comprised of four miRNAs (miR-98-p, miR-152-3p, miR-326 and miR-4289), which has the potential to improve the diagnostic power of the current PSA biomarker alone [26]. In order to determine if the upregulation in PCa samples observed by us in two cohorts is specific of this disease, we have carried out a systematic review, where the dysregulation of these four miRNAs in cancer has been thoroughly looked at in the literature. Additionally, a dedicated cancer miRNA dysregulation database has also been studied and, finally, information on association of miRNA expression level with survival has been compiled.

Our own analysis showed significantly higher expression plasma levels of miR-98-5p, miR-152-3p, miR-326 and miR-4289 in prostate cancer patients when compare to healthy individuals in two cohorts and this was associated with an increased risk of developing the disease for the first three miRNAs. Opposite results were obtained for miR-4289 between the two cohorts analysed. From our literature search, the upregulation of miR-98 and miR-326 in PCa suggests them to be PCa-specific when compared to other cancers. By contrast, miR-152 lacked specificity for this disease and not enough data for miR-4289 was available to draw any conclusions. Most studies are on tumor samples in PCa. Interestingly, the upregulation of miR-98 in recurrent PCa patients after radical prostatectomy when compared to non-recurrent patients was also reported [29], placing this miRNA not only as a potential diagnostic marker but also as a prognosis possible tool for this disease. To complement the information found in the literature, we then looked at the dedicated miRNA cancer dysregulation database, dbDEMC2, since most of the values reported in here have not been reported in any research articles. Unexpectedly, our analysis from this database did not show similar results to those observed in the literature. Instead, none of the miRNAs showed a specific dysregulation in any of the cancers studied. This could be due to the nature of the miRNA detection assay used (microarray) and small sample sizes. While the majority of the experimental data obtained from the literature used a specific amplification method of the miRNA of interest (quantitative reverse transcription polymerase chain reaction), the data obtained from the aforementioned database used a somewhat more general method of detection, the microarray approach. In this method, hundreds of probes are used to get an initial sense of possible expression data patterns. Parameters such as the type and length of the probes, how these are manufactured, and how many tissues/liquid samples in each study can affect the results. Survival data consistently showed high levels of miR-98, miR-152 and miR-326 were associated with a higher survival rate and low aggressive disease for several cancers except PCa, where an upregulation was associated with a poorer prognosis (HR > 1) and early biochemical recurrence. The research articles included in this review, also provided information about several validated targets that can explain the dysregulation of the miRNAs of interest in the cancers studied or the molecular consequences. Some of these targets are oncogenes, for example, CCND1 and KRAS, or tumour suppressors such as PTEN and HSP90AA1, and the deregulation of these genes may contribute to cancer pathogenesis.

A potential limitation of this study is the lack of a standardised miRNA extraction protocol from clinical samples available in the scientific community. As shown recently [78,79], detected levels of miRNA can significantly vary depending on the extraction methodology applied and the internal controls used, particularly in serum samples. Additionally, the pre-extraction steps such as sample collection, specimen acquisition, handling and storage can further impact the levels detected downstream [79]. A second limitation is the mixed origin of the samples. While we reported an upregulation in PCa plasma samples for the four miRNAs, most of the expression data found in both the literature and the microarray-based database was from tumor tissue samples. Additionally, miRNA concentration levels in serum than plasma have been observed due to unknown effects of coagulation process on extracellular miRNAs in blood [79], which makes drawing comparisons regarding miRNA dysregulations across different samples even more difficult. This could raise some concerns about how to interpret the data. For example, miR-152 was upregulated in serum from bladder cancer patients [80] when compared to controls while, in a different study, it was downregulated in bladder cancer tissue patients when compared to healthy individuals [53]. Also, despite our data, where an increase of this miRNA expression was observed in plasma samples in PCa patients [26], three independent studies done by Zhu, Theodore and Kristensen et al. respectively [38,39,40], observed a downregulation in PCa patients’ tissue when compared to adjacent or healthy tissue controls. However, it is important to note that the downregulation reported in the last study by Kristensen et al. compares PCa tissue with BPH tissue and, therefore, the results should be contrasted with caution. This inverse relationship in terms of miRNA expression between plasma and tumour may be due to the possibility of the preferential retaining of oncomirs (miRNAs overexpressed in cancer [81]), and the release of tumour suppressor miRNAs into the circulation to promote tumourigenesis. It has been previously proposed that the preferential export of certain microRNAs may occur as hormomirs (circulatory miRNAs) which may function to modulate expression of genes at a secondary site, thus affecting the pathology of the disease [82,83]. We could also hypothesise that the downregulation in one sample type (tissue) and the upregulation in the other (plasma) is due to the cancer physiological conditions, which causes a leakage of molecules, being this the basis of the PSA blood test for PCa [84]. To explain some of the downregulation observed in tissue by others, we took special interest in two potential mechanisms due to the high number of observations reported. It is well known that DNA methylation is increased during cancer [85,86] and the fact that the miR-152 promoter area is surrounded by a CpG island [87] makes this miRNA highly vulnerable to an aberrant regulation in this malignancy. In fact, the hyper-methylation in this region has been associated with a lower expression of miR-152 in cancer tissue, inhibiting its oncosuppressor properties. In particular, this was observed in prostate, ovarian, breast and glioma cancers [40,43,47,88]. Interestingly, this effect was shown to be ethnicity-associated in a PCa study [40], where an African American cohort presented a higher methylation rate in the miR-152 gene than a Caucasian cohort, and this was also associated with a more aggressive form of the disease. A very recent published study has also significantly associated a lower miR-152-3p expression in PCa tissue patients (*N* = 100) when compared to healthy tissue (*N* = 14) with a highly hyper-methylated promoter region [89], making this association a very plausible mechanism which should be further explored. Another interesting possible mechanistic explanation for the reported downregulation of miR-326 in cancer tissue when compared to healthy controls is the negative regulator long non-coding RNA HOTAIR (lnc-HOTAIR). Lnc-HOTAIR has been shown to negatively regulate this miRNA in lung and ovarian cancers [59,68] and its overexpression has been associated with a poor prognosis, metastasis promotion and carcinogenesis in several cancers [69,90].

In summary, cancer expression data has been reported for miR-98, miR-152 and miR-326. From the research articles included, the upregulation of miR-98 and miR-326 in PCa was shown to be highly specific. In most cases, these three miRNAs were downregulated in cancer and this was often associated with a poorer prognosis, later staging, increased tumor size, and higher pathological grades. We have reported an upregulation of the miRNA signature miR-98-5p, miR-152-3p, miR-326 and miR-4289 in PCa plasma patient samples when compared to healthy controls. Since our signature’s upregulation has been detected in plasma, and most of the dysregulated expression data comes from tissue samples, the results presented here should be taken as a summary of what data is available for these four miRNAs expressions in cancer to date. Indeed, further studies are needed to clarify their plasma and serum dysregulations in different cancers and their suitability as specific markers of PCa.

## 4. Materials and Methods

### 4.1. Research Data Extraction Strategy: Inclusion and Exclusion Criteria

The literature used within this study was obtained via the PubMed database, utilising studies conducted in the last 10 years and the strategy followed is summarised in Figure 2. Briefly, studies included had to be tested on at least 20 patients and involve clinical data such as disease pathological characteristics or clinical outcome. After an initial search and exclusion following the above criteria, more than 200 abstracts were pre-viewed. From these, 78 full publications were retrieved for the systematic review. Only those research articles that reported significant data were finally included. Thus a total of 38 studies, including our own, were incorporated for the dysregulated miRNA expression review (Table 1) and/or their prognostic potential (Table 2). Some of the selected studies presented expression data for >20 samples but their survival data was reported for <20 patients. This latter data has also been presented in Table 2. Some studies reported exact P values whereas others reported them as thresholds (re P < 0.05, P < 0.01, etc.) and these have been presented accordingly. Our own reported data has been reported by the regression coefficient (β value) with a 95% CI.

### 4.2. Differential Expression of the miRNA Signature in Cancer: High-Throughput Data

In addition to the reported data in the literature for this signature, we extracted normalised logarithmic fold change miRNA expression data from the dedicated Differentially Expressed miRNAs in human Cancers database (dbDEMC version 2.0, http://www.picb.ac.cn/dbDEMC) (data extracted in March 2018) [66]. This integrated database differs from others since it identifies these differentially expressed miRNAs from de novo analysis of high-throughput expression data. Additionally, it contains differential expression of 2,224 miRNAs from 36 cancers curated from 436 experiments. The current version displays data detected by high-throughput methods from a total of 209 newly published data sets, collected from GEO/TCGA. Expression values were logarithmically transformed (base 2) and quantile normalized. When different isoforms were available, dbDEMC selects the one with the greatest expression value. In Figure 1, cancer vs. normal comparison is represented for the miRNA signature, where the fold change values for each miRNA were plotted using GraphPad Prism 7.00. Each panel was subsequently analysed using an unpaired, non-parametric ANOVA test and a P < 0.05 was considered significant.

### 4.3. Dysregulation of miR-98-5p, miR-152-3p, miR-326 and miR-4289 Associated with Cancer Prognosis

Those studies that reported survival data and/or expression in metastatic tissue when compared to non-metastatic disease are summarised in Table 2. HR and 95% CI were included when available. HRs are used to measure survival rates through time when comparing more than one group of patients [91]. The aim is to determine if the survival outcome is different amongst the groups studied due to a particular variable that differentiates them such as treatments or, in our case, different levels of miRNAs. As seen with β values, a HR value >1 is considered to be associated with a poorer prognosis and shorter overall survival while an HR <1 is associated with a more favourable one. HRs were calculated by multivariate Cox binary logistic regression with P < 0.05 unless otherwise specified. Some studies reported their HRs based on the downregulation of the expression of the miRNA of interest. In order to keep the data homogeneous, in these cases, the inverse of both the ratios and their 95% CI were calculated (1/HR and 1/95% CI), allowing for the different studies to be compared more easily. These studies are marked using the superscript “ᵅ” in Table 2.

## 5. Conclusions

The upregulation of miR-98 and miR-326 in PCa were shown to be highly specific of prostate cancer as in the majority of the cases reported these two miRNAs were downregulated in cancer and they were often associated with a poorer prognosis that included later staging, larger tumor size, and higher pathological grades. The miRNA signature we have recently reported showed that miR-98-5p, miR-152-3p, miR-326 and miR-4289 were upregulated in PCa plasma patient samples when compared to healthy controls. A limitation of our study includes the nature of our samples (plasma) which differs from most of the samples’ origin found in the literature (tissue). Therefore, here we present a snapshot of the significant reported dysregulation in cancer of these four miRNAs to date.

## Figures and Tables

**Figure 1 ijms-20-01154-f001:**
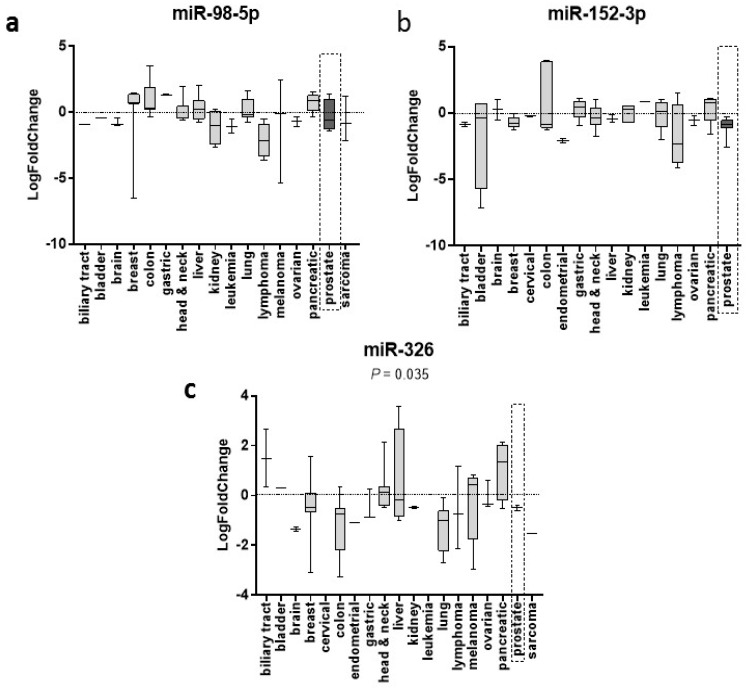
Differential expression of miR-98-5p, miR-152-3p and miR-326 in cancer tissue. Data sourced from dbDEMC [66]. Logarithmic fold changed plotted (GraphPad Prism 7.00) showing min to max values and median. Dark grey shade and dashed rectangular highlight PCa only. P values obtained from one-way analysis of variance (ANOVA).

**Figure 2 ijms-20-01154-f002:**
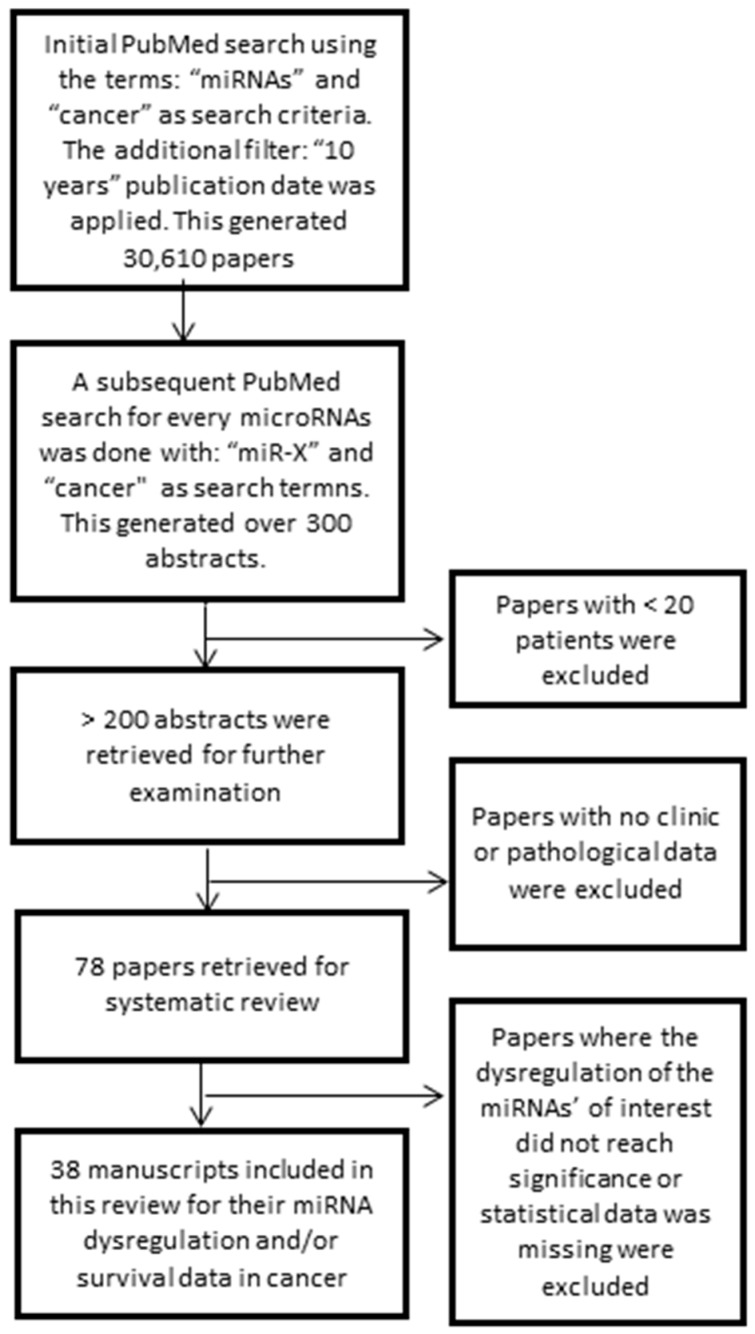
Diagram showing the overall pathway followed to obtain reported significant data for miR98, miR-152, miR-326 and miR-4289 deregulation in human cancers.

**Table 1 ijms-20-01154-t001:** Dysregulation of miR-98, miR-152, miR-326 and miR-4289 in prostate cancer (PCa) when compared to cancer-free samples (tissue, serum or plasma).

miRNA	Prostate	HCC	ESCC	Glioma	Lung	Breast	CRC	Bladder	Ovarian	Cervical	Gastric	Melanoma	NPC	OSC
miR-98	↑^P^ [26,29]ᵒ	↓^T^ [30,31]	↓^T^ [32]	↓^T^ [33]	↓^T^ [34] ↓^S^ [35]							↓^T^ [36]	↓^T^ [37]	
miR-152	↑^P^ [26] ↓^T^ [38]^#^ ^↓^^T^ [39,40]	↓^T^ [41,42]		↓^T^ [43]	↓^T^ [44] ↓^P^ [45] ↑^P^ [46]	↓^T^ [47,48] ↑^P^ [46]	↓^T^ [49,50,51] ↑^P^ [46]	↑^S^ [52]↓^T^ [53]	↓^T^ [54]		↓^T^ [50,55]			
miR-326	↑^P^ [26]				↓^T^ [56]		↓^T^ [57]			↓^T^ [58,59]	↓^T^ [60]			↓^T/S^ [61]
miR-4289	↑^P^ [26]													

HCC: hepatocellular carcinoma, ESCC: esophageal cancer, CRC: colorectal cancer, NPC: nasopharyngeal Cancer, OSC: osteosarcoma, Kristen et al.: ^#^ downregulation in benign prostatic hyperplasia PCa tissue. S: serum, T: tissue, P: plasma, ᵒPashaei et al.: recurrent vs. non-recurrent after radical prostatectomy patients, ↑: upregulated, ↓: downregulated.

**Table 2 ijms-20-01154-t002:** Dysregulation of miR-98-5p, miR-152-3p, miR-326 and miR-4289 associated with cancer prognosis.

miRNA	Cancer	Sample Type	Number of Samples	Hazard Ratio (HR) (95% Confidence Interval CI)	High %Survival	Mets. vs. No-Mets Tissue	Reference
miR-98-5p	Lung	Tissue	26	-	↑	↓	[35]
	Melanoma	Tissue	15	-	↑	↓	[36]
	HCC	Tissue	144	-	↑	↓	[30]
	ESCC	Tissue	40	-	-	↓	[32]
miR-152-3p	HCC	Serum	76	0.39 (0.19–0.82) ᵒ	↑	-	[42]
	Prostate	Tissue	13	-	↑*	↓	[40]
miR-326	Prostate	Tissue	126	1.1 (1.01–1.2) ᵒ^,^ᵞ	↓ᵞ	-	[38]
	Prostate	Tissue	110	1.1 (1.01–1.2) ᵒ^,^ᵞ	↓ᵞ	-	[38]
	CRC	Tissue	114	0.58 (0.3–0.8)	↑	↓	[61]
	GBM	Tissue	458	0.7 (0.5–0.9)	↑	-	[62]
	Glioma	Tissue	108	0.15 (0.05–0.8) ᵒ	↑	-	[63]
	OSC	Serum	60	0.25 (0.1–0.9) ᵒ	↑	↓	[61]
	Gastric	Tissue	136	0.7 (0.4–0.9) ᵒ	↑	↓	[60]
	Lung	Tissue	39	-	↑	-	[56]

HCC: hepatocellular carcinoma, ESCC: esophageal cancer, CRC: colorectal cancer, GBM: glioblastoma, OSC: osteosarcoma, Mets: metastatic, *: biochemical recurrence-free survival, ᵒ: for those studies that HR values were reported for low miRNA, the inverse HR has been calculated, ᵞ: not validated in a third cohort and early biochemical recurrence (two cohorts), ^#^: not validated in a multivariate analysis, ↑: upregulated, ↓: downregulated.

**Table 3 ijms-20-01154-t003:** Functional role of the miRNA signature in cancer.

miRNA	Cancer	Target Gene/Regulator	Method of Validation	Binding Target Effect	Pathway or Molecular Function	Reference
miR-98	Lung	Integrin Subunit Beta 3 (ITGB3)	Cancer cell line transfection and mouse injection model	Cancer cell proliferation suppression and tumor growth reduction in vivo	Cell adhesion and cell-surface mediated signalling	[34]
		Serine/threonine-protein kinase (PAK1)	Cancer cell line transfection and mouse injection model	Cancer cell proliferation, colony formation, migration and invasion inhibition	Cytoskeleton reorganization and nuclear signalling	[67]
	Melanoma	Interleukin 6 (IL-6)	Mouse injection model	Tumor metastasis and growth inhibition in vivo	Stat and NF-κB signalling	[36]
	HCC	Collagen Triple Helix Repeat Containing 1 (CTHRC1)	Western blot	Protein expression inhibition after mimics transfection	Tissue remodelling, vascularity and bone formation	[31]
	Brain	Raf-1 kinase inhibitor protein (RKIP)	Cancer cell line transfection	miR-98 positive regulator: tumor repressor	Inhibition of the Raf-1-MEK1/2, ERK1/2 and NF-kappaB signalling pathways	[33]
	NPC	Signal transducer and activator of transcription 3 (STAT3)	Western Blot	Protein expression inhibition after mimics transfection	Transcription factor	[37]
miR-152	Gastric	CD151	Cancer cell line transfection	Cancer cell proliferation, migration and invasion suppression	Tetraspasin member (cell development, growth and motility regulation)	[55]
	Ovarian	SERPINE1	In silico transcriptome analysis	Tumorigenesis and metastasis suppression	tPA/uPA fibrinolysis inhibitor	[68]
	Breast	Insulin-like growth factor 1 (IGF-IR)	Cancer cell line transfection	Cancer cell proliferation, colony formation and tumor angiogenesis inhibition	PI3K/AKT and MAPK/ERK cascades	[47]
		Insulin receptor substrate 1 (IRS1)	Cancer cell line transfection	Cancer cell proliferation, colony formation, and tumor angiogenesis inhibition	PI3K/AKT and MAPK/ERK cascades	[47]
		Rho-Associated Coiled-Coil Containing Protein Kinase 1 (ROCK1)	Cancer cell line transfection	Cancer cell proliferation, migration and invasion suppression	GTPase RhoA multiple signalling cascade	[48]
	Lung	Neuropilin-1 mediated receptor	Cancer cell line transfection	Cancer cell migration and invasion suppression	VEGF-A, VEGF-165	[54]
miR-326	Gastric	Fascin (FSCN1)	Cancer cell line transfection	Cancer cell growth and metastasis suppression	Formation of actin-based cellular protrusions	[60]
	Lung	Cyclin D1 (CCND1)	Cancer cell line transfection	Cancer cell proliferation, migration, invasion and colony formation inhibition and promotion of apoptosis	CDK kinases regulator coordination of each mitotic event	[56]
		Paired-like homeobox 2a (phox2a)	Cancer cell line transfection	Cancer cell proliferation, migration and invasion suppression	Development of the autonomic nervous system	[69]
		LncRNA-HOTAIR	Cancer cell line transfection	miR-326 negative regulator: promotion of cancer cells proliferation and migration	LSD1/PRC2 epigenetic regulator	[69]
	Glioma	Nin one binding protein (NOB1)	Cancer cell line transfection	Cancer cell proliferation and colony formation inhibition	Pre-rRNA processing and MAPK signalling	[70]
	OSC	B-cell lymphoma 2 (Bcl-2)	Cancer cell line transfection	Cancer cell apoptosis reduction	Caspase activity regulator	[61]
	Cervical	LncRNA-HOTAIR	Cancer cell line transfection	miR-326 negative regulator: promotion of cancer cells proliferation and migration	LSD1/PRC2 epigenetic regulator	[59]

HCC: hepatocellular carcinoma, NPC: nasopharyngeal carcinoma, OSC: osteosarcoma, tPA: tissue plasminogen activator, uPA: urokinase, phosphatidylinositol-3 kinase (PI3K)/AKT and mitogen-activated protein kinase (MAPK)/extracellular signal-regulated kinases (ERK) cascades, RhoA: Ras homolog gene family, member A, VEGF: vascular endothelial growth factor, CDK: cyclin-dependent kinases, LSD1: lysine specific demethylase 1, PRC2: polycomb repressive complex 2, Lnc: long non-coding, MAPK: mitogen-activated protein kinases.

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
