# Peer review of "Assessment of miR-98-5p, miR-152-3p, miR-326 and miR-4289 Expression as Biomarker for Prostate Cancer Diagnosis"

_ijms, 2019, doi:10.3390/ijms20051154_

Round 1
Reviewer 1 Report
The authors have correctly answered the reviewers' comments. Now, the manuscript is well structured and clear. So, in my opinion the manuscript is acceptable for publication.
Reviewer 2 Report
After changing the article to review, it should be accepted for publication.
Reviewer 3 Report
Authors have revised the manuscript carefully according to the reviewer's comments and suggestions.
This manuscript is a resubmission of an earlier submission. The following is a list of the peer review reports and author responses from that submission.
Round 1
Reviewer 1 Report
The authors reviewed the role of MiR-98-5p, -152-3p, -326 and -4289 as biomarkers for prostate cancer diagnosis, selected according to previous data published by the authors. These data were additionally complemented with data obtained from the dbDEMC2 database.
This is a valuable paper, which can be published with minor changes:
Line 88. HR is defined in line 461. The authors should defined HR in line 88.
dbDEMC appears as dbDMEC in lines 293 and 449.
Line 351. PrCa is PCa
Line 357. The sentence “from the, did not show” appears incomplete.
The discussion is well developed and the authors mention several reasons to explain opposite results (up- or downregulation) in different tumors, including biological and analytical reasons. In fact, the application of miRNAs in clinical practice is constricted by the lack of a standardized measurement protocol. In my opinion it is a significant point. I suggest to the authors to include some relevant references to remark this point:
Marzi MJ, Montani F, Carletti et al. Optimization and Standardization of Circulating MicroRNA Detection for Clinical Application: The miR-Test Case. Clin Chem. 2016 May;62(5):743-54
Filella X, Foj L. miRNAs as novel biomarkers in the management of prostate cancer. Clin Chem Lab Med. 2017 May 1;55(5):715-736
Reviewer 2 Report
To be honest, I don't think that this article qualifies as "original content", since it just discusses a potential miR pattern for PCa which the authors have established before in a rather small patient collective without external validation. The only original content i see here is assessing publicly availabe expression data of different tissues.
I would reconsider this if it were a review kind of article, but in its present form, it just does not qualify. Potential target genes (derived from other cancer entities) of the miRs presented are described, but a comprehensive validation in the form of qRT-PCR / WB / Dual Luciferase assays (except for qRT-PCR experiments in the original publications in scientific reports) in PCa lack completely to somehow underpin the claims made.
My suggestion: either add some fundamental experimental content (this would be educational) or re-submit as review.
Reviewer 3 Report
In this manuscript, authors have performed a literature review on a previously identified panel of four miRNAs (miR-98-5p, miR-152-3p, miR-326 and miR-4289) with elevated levels in plasma samples from prostate cancer (PCa) patients compared to healthy controls. They have reviewed over 200 research articles published on these four miRNAs in all reported cancers in the last decade. Further, they have studied differentially Expressed miRNAs in human Cancers database (dbDEMC) and obtained information on association of these four miRNAs expression level with patient survival. By reviewing the available literature, authors suggest that these miRNAs are dysregulated in various cancer types including PCa and may have potential for PCa specific diagnosis.
Overall, the manuscript presents some interesting information on a panel of four miRNAs, which may be used for further studies on these miRNAs in cancer. However, there are some corrections required that are provided below.
1. Authors may provide inclusion and exclusion criteria for studies reviewed in this study as a separate heading in materials and methods section.
2. In the legend to fig 2B, authors have written miR-153-3p. Please correct it to miR-152-3p.
3. In the legend to fig 2, authors should specify that the data presented in the figure 2 is mostly from cancer tissues or from both tissues and plasma.
4. Table 1 and 2, authors should mention what does arrow (up/down) indicate.
5. Authors are suggested to be consistent in writing and carefully proofread the manuscript for typing mistakes. For example, somewhere they have written PrCa but most of the places PCa.